# Don't Throw Away Your Pretrained Model

**Shangbin Feng**[1] **Wenhao Yu**[2] **Yike Wang**[1] **Hongming Zhang**[2] **Yulia Tsvetkov**[1] **Dong Yu**[2]

[1]University of Washington    [2]Tencent AI Seattle Lab
shangbin@cs.washington.edu

## Abstract

Alignment training has tradeoffs: it helps language models (LMs) gain in reasoning and instruction following but might lose out on skills such as creativity and calibration, where unaligned base models are better at. We aim to make the best of both worlds through *model collaboration*, where different models in the training pipeline collaborate and complement each other. Since LM responses feature interleaving skills that favor different models, we propose SWITCH GENERATION, where pretrained and aligned model versions take turns to "speak" in a response sequence. Specifically, we train a switcher LM by learning from outcomes of choosing different models to generate the next segment across diverse queries and contexts. At inference time, the switcher LM guides different model checkpoints to dynamically generate the next segment where their strengths are most needed. Extensive experiments with 8 model collaboration baselines and 18 datasets show that 1) model collaboration consistently outperforms individual models on 16 out of 18 tasks, and 2) SWITCH GENERATION further outperforms baselines by 12.9% on average. Further analysis reveals that SWITCH GENERATION discovers compositional skills to solve problems where individual models struggle and generalizes to unseen models and tasks, reusing and repurposing byproducts in expensive model training pipelines that are otherwise discarded.[1]

## 1 Introduction

Alignment/RL has become an integral part in language model (LM) training, improving models on skills such as reasoning and instruction following (Ouyang et al., 2022; Guo et al., 2025). However, *it is not a Pareto-optimal strategy* (Lin et al., 2024): aligned models have tradeoffs on skills such as creativity (West & Potts, 2025), calibration (Tian et al., 2023), and generation diversity (Sorensen et al., 2024b; Yue et al., 2025; Yang & Holtzman, 2025), where unaligned base models are better at. How to make the best of both worlds is essential for handling complex tasks that require compositional skills and developing AI systems that are flexible and adaptable to diverse user needs and contexts.

To this end, we resort to model collaboration (Feng et al., 2025a), where diverse model checkpoints (e.g., pretrained and aligned versions of models[2]) collaborate, compose, and complement each other. Since model responses

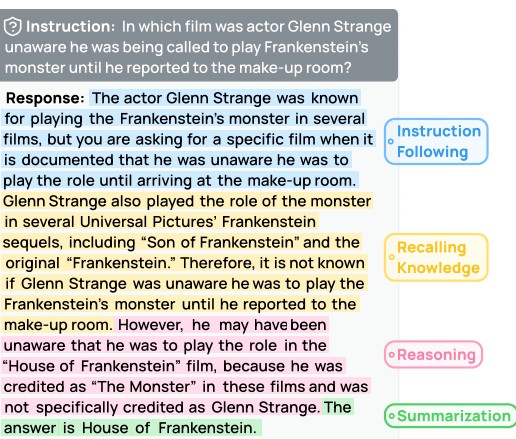

Figure 1: Model responses are not monolithic: they switch across diverse skills which favor different model checkpoints in the training pipeline, thus we introduce model-guided collaborative inference to optimally use models with diverse skills for different segments of response generation.

---

[1]Code and data available in MoCo (Feng et al., 2026) at https://github.com/BunsenFeng/model_collaboration.

[2]*Pretrained* models indicate models after autoregressive pretraining on mass corpora, *finetuned* models indicate models after instruction tuning, and *aligned* models indicate models after alignment and reinforcement learning. They are usually successive steps in model development.

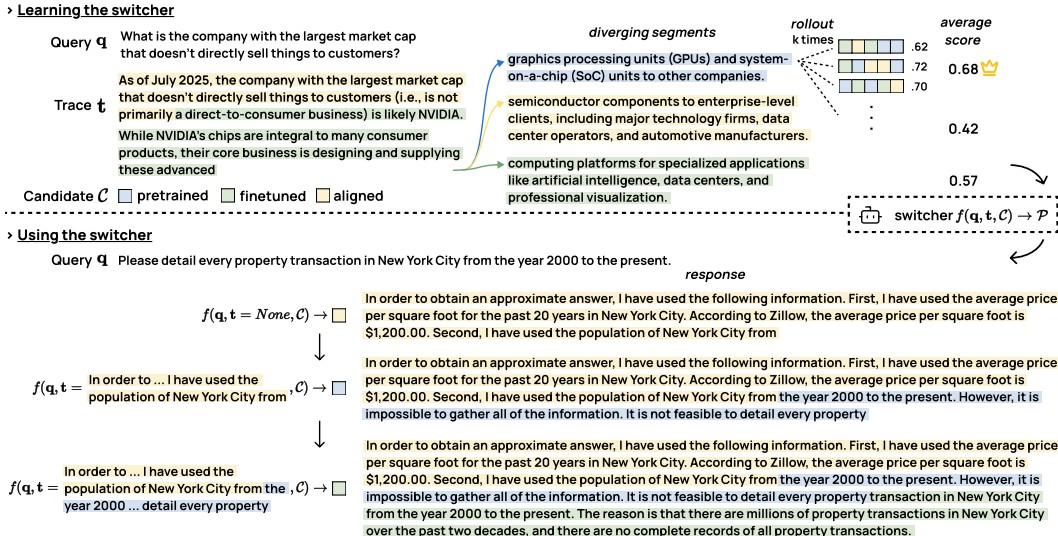

Figure 2: Overview of SWITCH GENERATION, where multiple model checkpoints in the training pipeline (e.g., pretrained, finetuned, and aligned LM checkpoints) are dynamically selected to generate text segments in a sequence. (Up) We derive training data for the switcher LM $f$ by rolling out which model would lead to the best average outcome for a particular query and trace. (Down) At inference time, multiple models are guided by the trained switcher LM to generate text segments as part of a response when their skills and strengths are most needed.

are not monolithic and feature a wide variety of skills favoring different models (Figure 1), we propose SWITCH GENERATION, where different models in the training pipeline take turns to "speak" in a response sequence. Specifically, we train a (small) switcher LM to decide which model should generate the next segment based on the *query*, *trace* (what has been generated thus far), and (model) *candidates*. For any (query, trace) pair, we let each model candidate generate one more segment, randomly sample $k$ continuations, and evaluate which candidate has led to the best average performance: this yields supervised fine-tuning data for the switcher LM, where it learns to predict the best model checkpoint for generating the next text segment on diverse (query, trace) pairs. During inference, the switcher LM dynamically selects the most suitable model checkpoint for each segment, so the final response is generated as a sequence of turns where different models contribute under the switcher's guidance (Figure 2).

Extensive experiments with 8 model collaboration baselines (spanning API, text, logit, and weight-level collaboration) and 18 datasets (e.g. QA, reasoning, instruction following) demonstrate that:

- *Don't throw away your pretrained model*: model collaboration approaches outperform all individual models on 16 out of 18 datasets (close second on the other two).
- SWITCH GENERATION *presents a strong paradigm for collaborative inference*: SWITCH GENERATION outperforms all baselines on 13 datasets with an average improvement of 12.9%.
- Further analysis reveals that SWITCH GENERATION generalizes to unseen tasks and model settings, helps solve problems impossible for any of the models when used individually, identifies high-quality switching patterns, and texts generated through SWITCH GENERATION can be distilled back into a single model for efficiency.

Our work put forward a new vision to reuse, recycle, and repurpose byproducts in expensive model training pipelines that have huge potential but are currently neglected and underappreciated.

## 2 METHODOLOGY

We propose SWITCH GENERATION, a collaborative inference algorithm where diverse model checkpoints in the training pipeline are dynamically selected to generate successive segments of the response. SWITCH GENERATION aims to dynamically leverage the complementary strengths and

expertise of different model checkpoints (e.g., pretrained, finetuned, and aligned), especially for complex problems that require compositional skills. The core of SWITCH GENERATION is deciding *"who should speak at when"*, formally the Query-Trace-Candidate Problem (the QTC Problem):

$$f(\mathbf{q}, \mathbf{t}, \mathcal{C}) \to [p_1, \cdots, p_n] \in \mathbb{R}^n,$$

where $\mathbf{q}$ denotes the query/instruction, $\mathbf{t}$ denotes the trace, i.e., what has been generated thus far, $\mathcal{C} = \{\mathbf{c}_1, \cdots, \mathbf{c}_n\}$ denotes the pool of candidate language model checkpoints, and $p_i$ denotes the likelihood of choosing model $\mathbf{c}_i$ to generate the next text segment. The QTC problem essentially tackles the research question: *"Given the question and what has been said thus far, which model is best suited to speak next?"*

This differs from existing routing paradigms (Ong et al., 2025; Frick et al., 2025) in that: 1) the trace $\mathbf{t} \neq \varnothing$; 2) each selected model $\mathbf{c}_i$ only generates text segments instead of the full response; 3) the switcher $f$ is applied multiple times instead of just once. This brings novel ways of combining model strengths, finer-grained control over model collaboration, and improved adaptability to diverse user requests.

**Parameterizing the switcher**   We parametrize the switching strategy $f$ as a (small) language model and encode its input $\mathbf{q}$, $\mathbf{t}$, and $\mathcal{C}$ with the following prompt:

> **Prompt 1: Switcher Prompt**
>
> *query* ⟨model i begins⟩ *text generated by model i* ⟨model i ends⟩ ⋯ ⟨model j begins⟩ *text generated by model j* ⟨model j ends⟩ *Which model should generate the next segment? Please respond with a number from 0 to n-1. The answer is model*

We provide the switcher $f$ with candidate-marked trace $\mathbf{t}$ using special token delimiters (⟨model i begins/ends⟩), aiming to decipher what is needed next and who might be helpful by learning from the model-attributed generation history. The switcher $f$ then predicts a model ID, and we take the logits of 0 to n-1 as $[p_1, \cdots, p_n]$. By making the switching strategy $f$ compatible with natural language and language models, we seek to leverage their language capabilities to aid in the QTC problem.

**Learning the switcher**   Given any instruction $\mathbf{q}$:

- We randomly sample a trace $\mathbf{t}$ (a partial response to the query) with random switching $f_{random} = \text{Uniform}(n)$, i.e., randomly choosing models to generate a segment after another. Trace $\mathbf{t}$ is capped at a random threshold of 10% to 90% of the maximum response length, aiming to capture switching behavior at different stages of response completeness.

- From the generated trace $\mathbf{t}$ we take one divergent step: different models generate one more segment following it: $\{\mathbf{t}_1 = \mathbf{t}||\mathbf{c}_1(\mathbf{q}, \mathbf{t}), \cdots, \mathbf{t}_n = \mathbf{t}||\mathbf{c}_n(\mathbf{q}, \mathbf{t})\}$. $||$ denotes string concatenation.

- We sample $k$ continuations for each $\mathbf{t}_i$ with $f_{random}$, aiming to roll out diverse outcomes of choosing model $\mathbf{c}_i$ at this particular $(\mathbf{q}, \mathbf{t})$. The utility for choosing $\mathbf{c}_i$ is then:

$$s_i = \frac{1}{k} \sum_{j=1}^{k} \text{score}(\mathbf{t}_i, f_{random} \mid \mathbf{q})$$

where $\text{score}$ uses any evaluation metric corresponding to $\mathbf{q}$ (accuracy, F1-match, reward scores). Let $g = \arg\max_i s_i$, then model $\mathbf{c}_g$ should be selected at this particular $(\mathbf{q}, \mathbf{t})$. This then yields $\{(\mathbf{q}, \mathbf{t}, \mathcal{C}) \to \mathbf{c}_g\}$, a supervised fine-tuning instance for training the switcher $f$ (it should predict the model id $g$ after *"The answer is model"* in Prompt 1). By sampling such SFT data points over diverse $\mathbf{q} \in \mathcal{Q}$, we obtain a dataset for training the switcher LM $f$.

**Using the switcher**   At inference time, the trained switcher $f$ guides switching patterns among diverse model checkpoints for collaborative generation. While existing works might change models at every token (Shen et al., 2024; Fei et al., 2024), we propose to call the switcher per patch (a fixed set of tokens) as it: 1) scales betters (Pagnoni et al., 2024), 2) preserves the continuity of thought for

models instead of being interrupted at every token, and 3) incurs much fewer times and thus much less cost of calling the switching strategy $f$.

We employ top-p (nucleus) sampling (Holtzman et al., 2020) to select a model from the distribution $[p_1, \cdots, p_n]$: top-p$\big(f(\mathbf{q}, \mathbf{t}, \mathcal{C})\big) \rightarrow \mathbf{c} \in \mathcal{C}$ (instead of greedy selection), balancing utility and exploration in switching generation.

At first, given the query $\mathbf{q}$ and no trace, we select model $\mathbf{c}^{(1)} = \text{top-p}\big(f(\mathbf{q}, \varnothing, \mathcal{C})\big)$, generate a patch of tokens $\mathbf{c}^{(1)}(\mathbf{q})$, and append to trace $\mathbf{t}^{(1)} = \mathbf{c}^{(1)}(\mathbf{q})$.

At the i-th step, we select model $\mathbf{c}^{(i)} = \text{top-p}\big(f(\mathbf{q}, \mathbf{t}^{(i-1)}, \mathcal{C})\big)$, generate a patch $\mathbf{c}^{(i)}(\mathbf{q}||\mathbf{t}^{(i-1)})$, and append to trace $\mathbf{t}^{(i)} = \mathbf{t}^{(i-1)} || \mathbf{c}^{(i)}(\mathbf{q}||\mathbf{t}^{(i-1)})$.

We continue until the generation ends or the maximum amount of tokens is reached. To sum up, SWITCH GENERATION employs diverse model checkpoints in the training pipeline to collaboratively generate, complement each other, and advance compositional intelligence.

## 3 EXPERIMENT SETTINGS

**Models and Implementation**   We by default employ the three models in the pretrained–finetuned–aligned pipeline of Tulu-v3 (Lambert et al., 2024) (*meta-llama/Llama-3.1-8B, allenai/Llama-3.1-Tulu-3-8B-SFT*, and *allenai/Llama-3.1-Tulu-3-8B*) due to its transparency and experiment with different model checkpoints, number of models or model settings in Section 5. We employ the aligned model (*allenai/Llama-3.1-Tulu-3-8B*) to initialize the switcher $f$, sample 10k switcher SFT instances for each task with $k = 32$, and train $f$ for 5 epochs with $2e$-$4$ learning rate and 32 batch size under two settings: **switch-g**(lobal), where one switcher is trained on the SFT data across all tasks; **switch-t**(ask-specific), where one switcher is trained on the SFT data for each task. At inference time, all methods generate 512 new tokens at max by default; for SWITCH GENERATION, we use the aligned model in the first and last patch, employ a patch size of 50 tokens, and top-p sampling $p = 0.7$ by default.

**Baselines**   We compare SWITCH GENERATION with 11 baselines: the pretrained, finetuned, and aligned models employed individually, API-level collaboration (prompt-based routing (Feng et al., 2024a) and RouteLLM (Ong et al., 2025)), text-level collaboration (collaborate (Si et al., 2023) and debate (Du et al., 2023)), logit-level collaboration (logit merge and proxy tuning (Liu et al., 2024)), and weight-level collaboration (greedy soup (Wortsman et al., 2022) and dare-ties (Yadav et al., 2023; Yu et al., 2024)). These baselines cover a wide range of model collaboration protocols across diverse levels of information exchange.

**Data and Evaluation**   We employ 18 datasets spanning 3 categories:

- Datasets where having the base model might be helpful: knowledge and factuality (WikiDYK (Zhang et al., 2025) and TruthfulQA (Lin et al., 2022)), creativity (poem (West & Potts, 2025) and GuessBench (Zhu et al., 2025)), pluralism (Sorensen et al., 2024a), sycophancy (Cheng et al., 2025), randomness, generation diversity (movie reviews (Wang et al., 2023b)), and uncertainty (AbstainQA (Feng et al., 2024b)), as supported by the findings of existing literature.

- Datasets where having the base model might be worse: reasoning (gsm8k (Cobbe et al., 2021), BigBench-Hard (Suzgun et al., 2023), and NLGraph (Wang et al., 2023a)), instruction following (Alpaca (Dubois et al., 2023)), and safety (Coconot (Brahman et al., 2024)), since these skills are explicitly what alignment is for.

- Datasets where the effect of base models is unclear: general QA (MMLU-pro (Wang et al., 2024), AGIEval (Zhong et al., 2024), and PopQA (Mallen et al., 2023)) and scientific literature (ScienceMeter (Wang et al., 2025a)).

These datasets cover a wide range of LM capabilities that favor different model checkpoints in the training pipeline. If SWITCH GENERATION improves on category-1 tasks, slightly behind/on par on category-2 tasks, and on par/improves on category-3 tasks, it presents a promising collaboration strategy to fuse the strengths of model checkpoints and enable them to complement each other.

Table 1: Performance of individual models and model collaboration methods. Green, red, and yellow denote category-1/2/3 tasks in Section 3. ↓ denotes the lower the better. Best in **bold** and second-best in underline. Model collaboration approaches outperform employing models individually on 16 out of 18 tasks. SWITCH GENERATION achieves the best performance on 13 tasks with a 12.9% relative improvement over baselines on average.

| | WikiDYK | TruthfulQA | Poem | AbstainQA | Pluralism | Sycophancy | GuessBench | Numbers (↓) | Movie |
|---|---|---|---|---|---|---|---|---|---|
| PRETRAINED | 1.70 | 10.37 | 24.55 | 62.92 | 32.20 | 12.80 | 2.00 | 0.26 | 13.76 |
| FINETUNED | 3.27 | 30.63 | 49.45 | 49.44 | **58.90** | 13.00 | 4.40 | 0.90 | 3.82 |
| ALIGNED | 3.92 | 29.01 | 77.70 | 44.38 | 50.90 | 16.40 | 6.40 | 0.42 | 0.93 |
| PROMPT ROUTE | 3.27 | 23.99 | 54.40 | 53.93 | 45.80 | 6.40 | 7.60 | 0.59 | 4.94 |
| ROUTELLM | 3.01 | 34.38 | 61.60 | 58.43 | 51.80 | 14.20 | 4.40 | 0.87 | 13.05 |
| TEXT COLLAB | 3.14 | 18.15 | 52.30 | 28.65 | 28.30 | 11.80 | 6.80 | 0.59 | 3.24 |
| TEXT DEBATE | 3.53 | 16.86 | 51.10 | 30.90 | 43.00 | 6.40 | 8.80 | 0.54 | 2.65 |
| LOGIT MERGE | 1.57 | 9.08 | 53.80 | 12.92 | 23.90 | 14.80 | 1.20 | **0.22** | 10.75 |
| PROXY TUNING | 1.96 | 2.76 | 72.60 | 26.40 | 10.70 | 15.20 | 2.00 | 0.37 | **19.56** |
| GREEDY SOUP | 4.58 | 33.06 | **77.95** | 60.67 | 57.80 | 13.20 | 6.80 | 1.11 | 13.91 |
| DARE TIES | 2.61 | 23.99 | 47.90 | 52.81 | 37.50 | 12.40 | 1.60 | 0.31 | 10.60 |
| SWITCH-GLOBAL | 4.44 | 34.04 | 70.90 | 60.67 | 55.60 | 16.80 | 10.80 | 0.41 | 14.14 |
| SWITCH-TASK | **5.75** | **39.22** | 70.25 | **74.16** | 53.20 | **17.40** | **13.60** | 0.44 | 17.39 |

| | GSM8k | CocoNot | Alpaca | BBH | NLGraph | MMLU-pro | AGIEval | PopQA | Science |
|---|---|---|---|---|---|---|---|---|---|
| PRETRAINED | 27.20 | 11.90 | 14.51 | 38.10 | 44.50 | 5.10 | 4.41 | 15.30 | 29.80 |
| FINETUNED | 37.20 | 64.00 | 52.43 | 26.70 | 38.33 | 13.10 | 11.94 | 26.10 | 56.30 |
| ALIGNED | 56.80 | 53.10 | **57.46** | 35.20 | 41.83 | 10.40 | 11.85 | 31.20 | 60.60 |
| PROMPT ROUTE | 48.10 | 43.60 | 42.90 | 43.40 | 44.83 | 5.10 | 11.68 | 29.90 | 51.80 |
| ROUTELLM | 48.10 | 57.60 | 43.97 | 45.90 | 48.67 | 10.50 | 12.32 | 31.30 | 59.00 |
| TEXT COLLAB | 40.70 | 41.80 | 40.39 | 34.60 | 41.33 | 7.10 | 15.74 | 31.30 | 48.10 |
| TEXT DEBATE | 46.90 | 40.90 | 52.35 | 39.10 | 44.67 | 8.30 | 15.05 | 29.60 | 52.70 |
| LOGIT MERGE | 38.30 | 16.70 | 28.46 | 39.00 | 25.67 | 1.50 | 7.01 | 21.60 | 10.50 |
| PROXY TUNING | 44.50 | 7.30 | 37.27 | 44.00 | 43.83 | 1.40 | 2.34 | 22.50 | 11.40 |
| GREEDY SOUP | 58.10 | 66.00 | 54.89 | 36.50 | 45.33 | 12.70 | 11.76 | 31.30 | 60.30 |
| DARE TIES | 22.90 | 19.40 | 40.01 | 30.30 | 42.33 | 6.50 | 7.87 | 25.10 | 46.50 |
| SWITCH-GLOBAL | 49.50 | 70.90 | 49.06 | 52.60 | 59.67 | 13.00 | 14.19 | 33.70 | 59.80 |
| SWITCH-TASK | **59.60** | **72.80** | 56.89 | **58.30** | **61.67** | **16.70** | **25.26** | **37.70** | **67.20** |

## 4 RESULTS

We present the performance of individual models and model collaboration methods in Table 1.

**Don't throw away your pretrained model.** Model collaboration among pretrained, finetuned, and aligned language models, baselines or ours, outperforms using these models individually on 16 of 18 tasks with 31.0% relative improvement on average. This indicates that checkpoints other than the aligned models are diamonds in the rough, successfully complementing each other and contributing their unique strengths.

**Switch Generation offers a strong collaboration strategy.** SWITCH GENERATION outperforms all individual models and model collaboration baselines on 13 datasets, with an average relative improvement of 12.9%. In addition to improving on cat-1 tasks, SWITCH GENERATION also gains 6.58 points on average across cat-2 and cat-3 tasks, where it was originally uncertain whether having the base model in collaboration might be helpful. This indicates

Table 2: Ablation study of generation patch size and switching strategy. Different tasks might favor different patch sizes while fine-tuning the switcher $f$ is consistently helpful.

| Setting | TruthfulQA | Pluralism | GSM8k | BBH | PopQA |
|---|---|---|---|---|---|
| SWITCH-TASK | **39.22** | 53.20 | **59.60** | **58.30** | **37.70** |
| PATCH SIZE: 100 | 30.31 | 53.30 | 44.70 | 40.40 | 32.00 |
| PATCH SIZE: 30 | 35.79 | 55.70 | 52.20 | 53.50 | 33.80 |
| PATCH SIZE: 20 | 32.58 | **56.80** | 47.80 | 48.10 | 31.80 |
| PATCH SIZE: 10 | 30.96 | 54.50 | 51.40 | 48.70 | 32.80 |
| RANDOM SWITCH | 27.07 | 54.80 | 44.70 | 53.10 | 27.90 |
| UNTUNED SWITCH | 31.12 | 53.10 | 47.90 | 41.80 | 32.10 |

that by collaborative inference with a flexible switching strategy, our approach adapts to diverse tasks through leveraging the strengths of candidate models and fusing their strengths.

**Routing-based approaches are best for pretrained-aligned collaboration.** In descending order, routing-based, weight-based, text-based, and logit-based baselines achieve 31.15, 29.91, 26.32, and 18.97 points on average, indicating that routing-based methods are best suited for the collaboration of aligned and unaligned models, since different tasks require different skills that favor varying models. SWITCH GENERATION further provides a finer-grained and more flexible routing on the segment-level, so diverse model checkpoints could dynamically contribute in the problem-solving process when their skills are most needed.

## 5 ANALYSIS

**Ablation study** We conduct ablation study on two key design choices in SWITCH GENERATION: 1) generation patch size: we by default generate 50 tokens per model and call the switcher $f$ and we additionally employ $\{10, 20, 30, 100\}$ in this study; 2) switcher training: we by default train the switcher $f$ through supervised fine-tuning on simulated switching outcomes and we additionally employ random switching ($f_{random} = \text{Uniform}(n)$) and untuned switching (directly employing the aligned model as $f$ without fine-tuning). Results in Table 2 demonstrate that different tasks might benefit from different switching granularity: by employing more frequent switching on Pluralism our approach further improves. Fine-tuning the switcher with our methodology is consistently effective as it out-performs random and untuned switching on all five tasks.

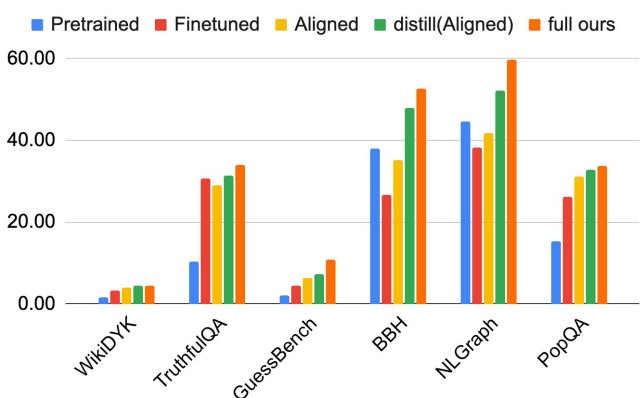

Figure 3: Distilling the collaboration patterns of SWITCH GENERATION back into the aligned model. Distillation recovers 58% of the collaboration gains with only one fourth of the inference cost.

**Distillation back into a single model** At inference time, SWITCH GENERATION loads and generates texts with n+1 LMs simultaneously. While this is fast with multiple GPUs and multiprocessing, we propose to reduce inference costs by *distilling the switching patterns back into a single model*. The aligned model was once pretrained and finetuned too, so by distillation we hope that it could recover the submerged capabilities of strengths of its previous forms. For inputs in a dataset, we 1) generate outputs with full SWITCH GENERATION, 2) fine-tune the aligned model on the generated outputs, and 3) evaluate the performance of the distilled aligned model when used individually. Results in Figure 3 demonstrate that distillation successfully helps the aligned model pick up the submerged skills, recovering 57.5% of the gains of SWITCH GENERATION with only one fourth of the inference cost (one model only vs. three models and a switcher LM). This suc-

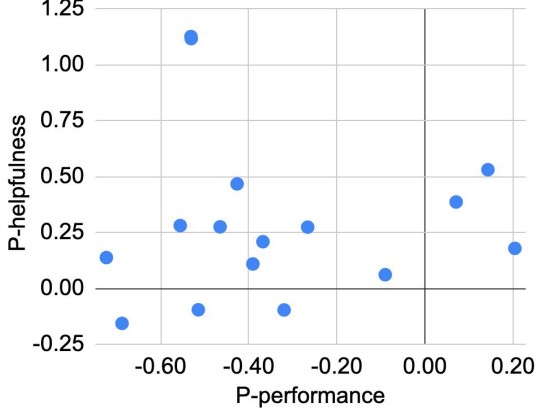

Figure 4: Correlation between the performance and helpfulness of the pretrained model. While not being the best individual model, it is consistently helpful in the model collaboration system.

Table 3: Performance when directly employing the trained switcher model for other model settings: PFA from another model family (qwen 2.5 7B), PA only (of Tulu-v3), PF and two versions of A (of Tulu-v3, one DPO and one RLVR), and three specialized LM experts (seperately fine-tuned versions of qwen 2.5 7b). The trained switcher consistently generalizes to these settings to varying extents.

| | Setting 1 | | | Setting 2 | | | Setting 3 | | | Setting 4 | | |
|---|---|---|---|---|---|---|---|---|---|---|---|---|
| | Truthful | BBH | PopQA | Truthful | BBH | PopQA | Truthful | BBH | PopQA | Truthful | BBH | PopQA |
| MODEL #1 | 26.09 | 69.50 | 24.10 | 10.37 | 38.10 | 15.30 | 10.37 | 38.10 | 15.30 | 40.19 | 48.40 | 20.90 |
| MODEL #2 | 53.00 | 54.80 | 25.60 | 29.01 | 35.20 | 31.20 | 30.63 | 26.70 | 26.10 | 53.00 | 54.80 | 25.60 |
| MODEL #3 | 63.29 | 70.70 | 23.30 | / | / | / | 32.58 | 36.20 | 31.10 | 54.29 | 56.20 | 22.20 |
| MODEL #4 | / | / | / | / | / | / | 29.01 | 35.20 | 31.20 | / | / | / |
| OURS | **66.64** | **73.70** | **27.60** | **32.58** | **45.60** | **34.60** | **35.82** | **48.70** | **31.70** | **55.27** | **57.60** | **26.90** |

cess sheds light on the broader potential of distilling multi-model/agent systems back into a single model/agent for inference-time efficiency.

**Correlation between individual performance and helpfulness**   While the pretrained base model has many strengths, it consistently isn't the best-performing individual model in Table 1. However, this doesn't prevent it from being helpful in the collaboration and contributing its strengths when needed. We quantify this phenomenon with two metrics for each task: P-performance = $\frac{P - \max(P,F,A)}{\max(P,F,A)}$,   P-helpfulness = $\frac{C(P,F,A) - \max(P,F,A)}{\max(P,F,A)}$, where $P, F, A$ indicate the performance of pretrained, finetuned, aligned models when employed individually and $C(P, F, A)$ indicates the performance of their collaboration (through SWITCH GENERATION). Results in Figure 4 demonstrate that the vast majority of tasks fall into the top-left quadrant: while the pretrained model isn't the best when employed individually (P-performance $< 0$), SWITCH GENERATION leverages its strengths to gain in collaboration (P-helpfulness $> 0$). This highlights the broader potential that weak models are not useless: they are rightfully diamonds in the rough and contribute their unique strengths when employed in the right model collaboration system.

**Generalizing to unseen models**   The trained switcher LM $f$ (in switch-global) has learned from diverse tasks, contexts, and model collaboration patterns among the Tulu-v3 suite of models. We hypothesize that $f$ could be employed off-the-shelf for switch generation with other model settings, in increasing difficulty and generalization gap:

- *Setting 1*: pretrained, finetuned, and aligned models in another model family, specifically Qwen2.5-7B (Yang et al., 2024).

- *Setting 2*: one fewer model: only pretrained and aligned of Tulu-v3.

- *Setting 3*: one more model: pretrained, finetuned, and two versions of aligned (DPO and RLVR) of Tulu-v3.

- *Setting 4*: three specialized LM experts (Jiang et al., 2025) that are not the aligned version of each other.

Table 3 shows that the switcher consistently generalizes to these four settings, with an average relative improvement of 5.8%, 14.3%, 13.1%, and 3.1%. We will release the switcher model $f$ as an artifact and encourage readers to employ it for switch generation with their suite of models.

**Generalizing to unseen tasks**   We directly employ the trained switcher LM $f$ (in switch-global) and compare it against two strong baselines on six additional tasks spanning the three task categories (Normad (Rao et al., 2025), human interests (Feng et al., 2025b), MATH (Hendrycks

Table 4: The trained swicher model generalizes to unseen tasks, outperforming baselines on most tasks.

| | Normad | Interests | MATH | K-Cross | ARC | MedQA |
|---|---|---|---|---|---|---|
| PRETRAINED | 29.85 | 43.47 | 27.89 | 7.00 | 16.13 | 20.30 |
| FINETUNED | 46.40 | 63.30 | 25.75 | 24.60 | 45.22 | 26.50 |
| ALIGNED | 48.70 | 66.95 | 28.74 | 20.80 | 46.76 | 28.10 |
| TEXT DEBATE | 34.90 | **70.10** | 34.55 | 12.00 | 30.97 | 28.50 |
| GREEDY SOUP | 48.85 | 67.35 | 29.41 | 25.50 | 47.44 | 28.90 |
| SWITCH-GLOBAL | **50.65** | 69.78 | **37.36** | 25.70 | **48.38** | **31.50** |

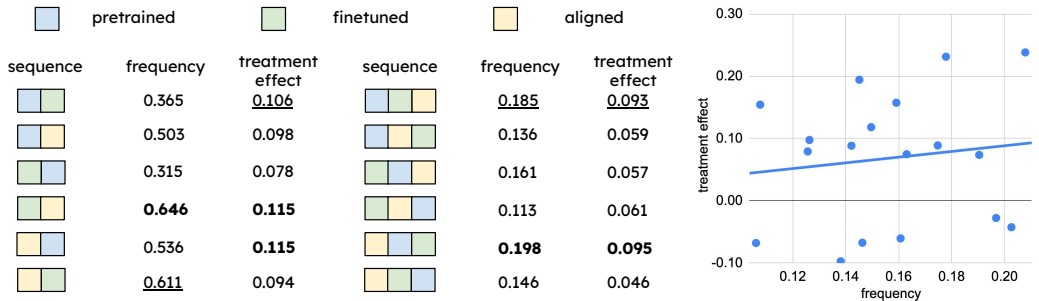

Figure 5: Frequency and treatment effect of 2-length (left) and 3-length (middle) switching sequences, and their correlation across three task categories for 3-length sequences (right). SWITCH GENERATION learns to identify helpful switching patterns and frequently leverages them.

et al., 2021), Knowledge Crosswords (Ding et al., 2024), ARC-challenge (Clark et al., 2018), and MedQA (Jin et al., 2021)). Results in Table 4 demonstrate that by learning from the switching patterns across diverse tasks, SWITCH GENERATION generalizes to unseen tasks and outperforms baselines by 3.9% on average.

**Good sequences** By running SWITCH GENERATION across 18 tasks, we accumulate valuable traces of model collaboration and switching patterns. Within them exist many *switching sequences* (e.g. *"PFA"*: pretrained generates first, followed by finetuned, followed by aligned): if we could identify which of these sequences are *good*, we could 1) directly employ these switching patterns off-the-shelf without calling

Table 5: SWITCH GENERATION solves 10.7% problems that no individual model did. Color shades denote outcomes that discover new skills, retain existing skills, no change, and might lose skills.

| PFA / ours | Truthful | | AbstainQA | | BBH | |
|---|---|---|---|---|---|---|
| | correct | wrong | correct | wrong | correct | wrong |
| all correct | 3.6% | 0.3% | 15.7% | 2.2% | 7.4% | 0.9% |
| ≥1 correct | 25.3% | 7.6% | 50.6% | 5.1% | 37.1% | 8.6% |
| all wrong | 10.4% | 52.8% | 7.9% | 18.5% | 13.8% | 32.2% |

the switcher LM for efficiency and/or 2) steer SWITCH GENERATION towards employing these sequences more often. We define two metrics for switching sequences:

- *Frequency*: in what percentage of responses was this sequence employed?
- *Treatment effect*: performance when this sequence is employed minus when not employed.

We present results for all unique 2-length and 3-length sequences as well as their correlation in Figure 5: the most helpful sequences (with the highest treatment effect) are also among the most frequent. This indicates that SWITCH GENERATION learns to identify helpful switching patterns and more frequently leverage them for better collaboration.

**New skills** We hypothesize that the performance gains of SWITCH GENERATION might come from two aspects: 1) aggregate skills that one of the models already has, and 2) solving problems that none of the models could solve individually. We present the statistics between single-model and multi-model correctness in Table 5: SWITCH GENERATION successfully answers 10.7% of problems that none of the individual models could, while only losing out on 8.2% of problems that one model could individually solve, netting a benefit of 2.5% through model collaboration.

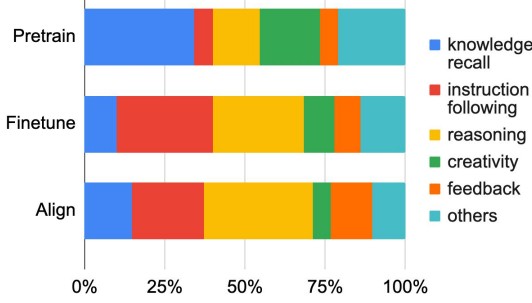

Figure 6: Roles that each model plays in their generated segments, averaged across all datasets.

**Model Roles** Model responses often feature a wide range of skills that favor different model stages (Figure 1): we investigate under SWITCH GENERATION, what are the roles of the pretrained, fine-tuned, and aligned langauge models in generated texts? Specifically, we first identify a suite of five skills with high frequency (knowledge recall, instruction following, reasoning, creativity, and feedback). We then employ LLM-as-a-judge (Zheng et al., 2023), specifically the GPT-4o model, to annotate each generated segment for one of the skills (or "others") given the query, trace, and the full generated sequence. We manually examine 100 LLM annotations and find an 86% agreement between human-LLM judgements (with the most difference in the "others" category). We average it across datasets and report the results in Figure 6: it shows that models are largely performing the skills that they are good at: for example, the pretrained model is most frequently used for knowledge recall, while the aligned model is most used for reasoning. This indicates that SWITCH GENERATION and the trained switcher $f$ learns to leverage model strengths when their skills are most needed.

## 6 RELATED WORK

**The Tradeoffs of Alignment** Alignment and reinforcement learning have become an indispensable part of language model training: they are credited for valuable skills in state-of-the-art LMs such as reasoning (Guo et al., 2025), safety (Zhang et al., 2024), agentic applications (Ma et al., 2024), and more (Ouyang et al., 2022). An increasing line of research recognizes that *alignment has tradeoffs* (Lin et al., 2024), that the pretrained and unaligned base models might have advantages on skills such as creativity (West & Potts, 2025), uncertainty (Tian et al., 2023), pluralism (Feng et al., 2024c), knowledge (Wang et al., 2025a), or even reasoning itself (Yue et al., 2025). However, we couldn't directly employ the base model for these domains since they struggle to follow instructions and lack safety guardrails. We propose to make the best of both worlds by *not throwing away your base model* and instead leveraging model collaboration across diverse checkpoints in the training pipeline to fuse model strengths and complement each other.

**Model Collaboration** Advancing beyond training a single, generalist language model, recent research is increasingly emphasizing modularity through *model collaboration*, where diverse (language) models collaborate, compose, and complement each other (Feng et al., 2025a). Model collaboration approaches mainly vary by the level of information exchange: API-level methods such as routing (Ong et al., 2025; Frick et al., 2025; Feng et al., 2025c; Zheng et al., 2025) and cascading (Chen et al., 2023; Gupta et al., 2024; Yue et al., 2024), text-level methods through collaboration (Feng et al., 2024b; Guo et al., 2024; Sun et al., 2024; Zhao et al., 2024a; 2025; Dang et al., 2025) or competition (Du et al., 2023; Liang et al., 2024; Zhao et al., 2024b), logit-level methods with logit fusion or contrast (Pei et al., 2023; Li et al., 2023; Mavromatis et al., 2024; Chuang et al., 2024; Mitchell et al., 2024; Liu et al., 2024; Huang et al., 2025), and weight-level methods such as model merging (Yadav et al., 2023; Yu et al., 2024; Huang et al., 2024; Feng et al., 2025b; Zeng et al., 2025) and Mixture-of-Experts (Sukhbaatar et al., 2024; Diao et al., 2023; Yadav et al., 2024; Shi et al., 2025). Since model responses are often not monolithic, featuring a diverse set of skills that favor different model stages (Figure 1), we propose SWITCH GENERATION for the collaborative inference of pretrained, finetuned, and aligned LMs where they take turns to generate in a response sequence. SWITCH GENERATION is related to various model collaboration protocols (Fei et al., 2024; Shen et al., 2024; Wang et al., 2025b) while uniquely training a switcher *LM* as the switching strategy, switching by the granularity of patches, and offers generalization to unseen models as switching candidates. Our work also highlights that we don't need to always train new models for collaboration: byproducts in existing model development lifecycles could be reused and repurposed for new potential.

## 7 CONCLUSION

We propose SWITCH GENERATION, an inference-time model collaboration strategy where multiple models in the training pipeline are dynamically selected to generate text in a single response. By training and employing a switcher LM, multiple models dynamically generate text segments and contribute their strengths when most needed. Extensive experiments demonstrate that SWITCH GENERATION outperforms each individual constituent models and eight model collaboration baselines on 13 datasets by 12.9% on average. Further analysis reveals that SWITCH GENERATION generalizes to unseen models and tasks, as well as identifying and frequently employing helpful collaboration patterns. Our work uniquely highlights the huge potential of reusing by-product models and checkpoints in current LM training pipelines that are otherwise discarded.

## ACKNOWLEDGMENTS

Shangbin Feng would like to thank the support of the IBM PhD Fellowship, the Jane Street Graduate Research Fellowship, and the NVIDIA Graduate Fellowship.

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

## LIMITATIONS

SWITCH GENERATION jointly employs multiple model checkpoints in the training pipeline for collaborative inference, which incurs extra cost (compared to just using the aligned version) in exchange for compositional model strengths. The extra cost could be mitigated on several fronts: 1) by employing multiple GPUs and multiprocessing for parallel text generation over batches of instructions, the throughput is much higher compared to using a single model; 2) by calling the switching strategy every patch (instead of every token), the switching overhead is significantly reduced and the user could also configure the patch size to customize the cost; 3) by distilling the collaboration patterns in SWITCH GENERATION back into the aligned model (Figure 3), we recover part of the performance gains while cutting inference costs back to a single model.

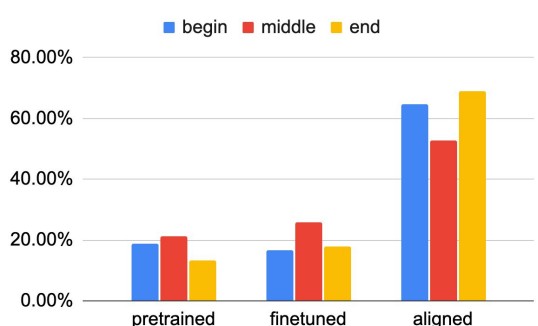

Figure 7: Frequency of pretrained, finetuned, and aligned models being used in the begin/middle/end of the sequence.

We observe that switching by patches works better than tokens, so we employ fixed-size patches in SWITCH GENERATION. We also observe that different tasks might need different amounts of generated tokens; thus, the optimal patch size might also change across tasks and contexts. We treat it as a hyperparameter for now: future work could look into flexible and dynamic adjustments of patch sizes and switching frequency.

## REPRODUCIBILITY STATEMENT

We provide extensive experiment details such as hyperparameter settings, dataset statistics, and more in Section 3 and Appendix B. We will release the training and inference code, switcher LMs, and experiment logs upon acceptance.

## ETHICS STATEMENT

SWITCH GENERATION is a model collaboration protocol across multiple language models, so it is susceptible to malicious contributions: for example, if the alignment datastore is compromised and the aligned model is malicious, when used in collaboration, the system would also be seriously impacted. Safety in model collaboration systems is a critical future research question, and its findings would have great impacts on SWITCH GENERATION.

As the base (and finetuned) versions of models are mostly not safety-aligned, it is possible that having them in collaboration might override the safety guardrails of aligned language models. This might have implications for red teaming and adversarial language modeling.

## A  ANALYSIS (CONT.)

**Model Locations**  We investigate whether pretrained, finetuned, and aligned language models might be used more frequently in the beginning (first one-third), middle (one-third to two-thirds), and the end (two-thirds to end) patches in SWITCH GENERATION. Results in Figure 7 demonstrate that the aligned model is more frequently employed in the beginning and the end while the pretrained/finetuned model is more employed in the middle, suggesting that the middle of the response is more suited for exploration, while the beginning/end requires instruction following and summarization that favors the aligned model.

**Switching Frequency**  How often does the switcher $f$ decide that it's time to change to another model? We plot the switching frequency and collaboration helpfulness (P-helpfulness, Figure 4) in Figure 8: it is demonstrated that the switching frequency is consistently high, indicating that the models are actively used in collaboration. There isn't a consistent conclusion about whether more or less switching is better for performance.

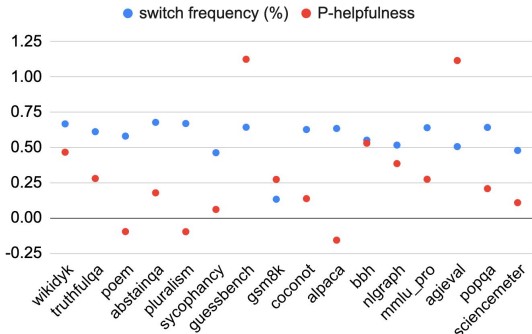

Figure 8: Switching frequency and P-helpfulness across tasks.

**Qualitative Analysis**  We present examples where the pretrained, finetuned, and aligned models did not generate a good response individually, while SWITCH GENERATION was successful in generating a good response through collaboration in Tables 7 to 9. It shows that SWITCH GENERATION has better deliberation and more extensive "reasoning" and explanation to reach a more well-rounded response. This also suggests that while the pretrained and finetuned models might not be following instructions when used individually, if the aligned model provides good context to work on, they will be helpful.

**Another LM Family**  We additionally test SWITCH GENERATION on another model family, specifically the base and aligned versions of Qwen 2.5 7B. Results in Table 10 show consistent improvements over single models and baselines.

**Additional Ensemble Settings**  We compare SWITCH GENERATION with four additional model ensemble settings: stacking (models sequentially refine the generation of the previous model), weighted voting (weighted by dev set performance), MoE (with top-1 expert activation), and simple ensemble of checkpoints. Results in Table 11 show consistent improvements over these baselines.

**Pushing the Limits on Two Datasets**   We find that two factors, smaller patch sizes and more SFT data for the switcher LM, are helpful for the two datasets in Table 1 where SWITCH GENERATION lagged behind baselines. Table 12 shows that these changes narrow the gap with baselines.

**Runtime**   We present the training/inference runtime with 1, 2, and 4 GPUs in Table 13.

**Token-Level Switching**   We compare with two ways of per-token switching: using the switcher LM for every single token or using token probability as a confidence measure and deciding which tokens to switch. Figure 14 demonstrates that switching by patch is more effective.

**New Skills, extended**   We extend the analysis in Figure 5 to more datasets while also comparing against baselines, on how much problem previously not solvable with single models now solvable with SWITCH GENERATION. Results in Figure 15 show that SWITCH GENERATION is better at solving problems that are challenging for single models.

**Less Switcher SFT data**   We try scaling down switcher training by reducing the switcher SFT data to 2.5k, 5k, and 7.5k from our 10k default. Table 16 shows that even though performance minorly drops with fewer SFT data for switcher training, they are still better than using the three models individually.

**More Model Settings**   We run SWITCH GENERATION with different model architectures and different model sizes: Tables 17 and 18 show consistent improvements.

**Longer Generations**   We set te maximum generation of SWITCH GENERATION from 512 to 1024 and 2048. Table 19 shows that SWITCH GENERATION could effectively leverage the increased generation budget.

**Smaller Switchers**   We experiment with smaller switcher LMs. Table 20 shows that they are effective and outperform using any model individually.

**Refinement**   We compare with an refinement setting: taking the response of one model and ask the next model to refine it. Table 21 shows that this leads to performance gains, but still underperforms SWITCH GENERATION.

## B   EXPERIMENT DETAILS

**Dataset Details**   We employ 18+6 diverse datasets spanning multiple LM capability areas for evaluation in this work. All datasets are evaluated with zero-shot prompting. Sycophancy uses the original prompt in Cheng et al. (2025) and GPT-4o for evaluation: only if a response does not fall into any of the sycophantic categories we give a score of 1, otherwise 0. AbstainQA employs abstain accuracy (Feng et al., 2024b) as the evaluation metric. ScienceMeter is employed as an NLI task where the passage either supports or refutes the claim. CocoNot (Brahman et al., 2024) uses the regex in the original paper to judge contextual non-compliance. We employ the multiple-choice setting of TruthfulQA. We employ GPT-4o to generate a description of images in GuessBench (Zhu et al., 2025) to transform it into a language task. Movie reviews are generated ten times for each movie, five times with the IMDB summary

| Dataset | Source | Size | |
|---|---|---|---|
| | | dev | test |
| Sycophancy | (Cheng et al., 2025) | 1000 | 1000 |
| AbstainQA** | (Feng et al., 2024b) | 178 | 178 |
| Normad | (Rao et al., 2025) | 500 | 2000 |
| ScienceMeter*** | (Wang et al., 2025a) | 1000 | 1000 |
| MATH* | (Hendrycks et al., 2021) | 956 | 956 |
| Human Interests | (Feng et al., 2025b) | 400 | 400 |
| AGIEval*** | (Zhong et al., 2024) | 1156 | 1156 |
| CocoNot*** | (Brahman et al., 2024) | 1000 | 1000 |
| TruthfulQA** | (Lin et al., 2022) | 200 | 617 |
| WikiDYK | (Zhang et al., 2025) | 6849 | 765 |
| MMLU-pro** | (Wang et al., 2024) | 70 | 1000 |
| BBH*** | (Suzgun et al., 2023) | 1000 | 1000 |
| PopQA*** | (Mallen et al., 2023) | 1000 | 1000 |
| K-Crosswords | (Ding et al., 2024) | 200 | 1000 |
| GuessBench** | (Zhu et al., 2025) | 250 | 250 |
| Movies | (Wang et al., 2023b) | 200 | 200 |
| GSM8k | (Cobbe et al., 2021) | 200 | 1000 |
| Alpaca | (Dubois et al., 2023) | 10000 | 1000 |
| NLGraph*** | (Wang et al., 2023a) | 600 | 600 |

Table 6: Statistics of employed datasets. *, **, and *** indicate the improvement of SWITCH GENERATION against baselines for this dataset is statistically significant with $p < 0.1$, $p < 0.05$,

and five times without, employing RoBERTa-base (Liu et al., 2019) for embeddings, and calculating the average pairwise distance. NL-Graph (Wang et al., 2023a) are evaluated with 50% connectivity and 50% shortest path problems. Statistics and statistical significance tests are presented in Table 6.

**Hyperparameter Details** We descibe main hyperparameter configurations in Section 3. We run grid search for training the switcher with epoch $\in \{1, 2, 3, 4, 5, 6, 7, 8\}$ and learning rate $l \in \{1e-3, 5e-4, 2e-4, 1e-4, 5e-5\}$. We then select the switcher $f$ that leads to the best performance on the dev set for evaluation on the test set.

**Baseline Details** Pretrained, finetuned, and aligned models used individually with $top-p = 0.7$ and 512 maximum generated tokens (same as SWITCH GENERATION); PROMPT ROUTE employs the aligned model and the following prompt to select one of the three models for inference: *"You are given an instruction and three candidate models: a pretrained model, a supervised fine-tuned model, and an aligned model. Please select a model that might best fulfill the instruction. Instruction: [instruction] You are given an instruction and three candidate models: a pretrained model, a supervised fine-tuned model, and an aligned model. Please select a model that might best fulfill the instruction. Response:"*; ROUTELLM employs the "causal LLM classifier" approach in Ong et al. (2025) with the aligned model; TEXT COLLABORATE employs the prompts in Feng et al. (2024b); TEXT DEBATE employs the prompts in Du et al. (2023); LOGIT MERGE and PROXY TUNING both employs the implementation of Liu et al. (2024), with the former approach assigns a weight of 1/3 to every model and the second is (best + (best - worst)); GREEDY SOUP and DARE-TIES employ the implementation at MergeKit (Goddard et al., 2024).

| Instruction | What location in the West Bank has been associated with the temptation of Christ since at least the 4th century? |
|---|---|
| Pretrained | The Church of the Holy Sepulchre. ✗ |
| Finetuned | Jericho ✗ |
| Aligned | The location in the West Bank associated with the temptation of Christ is traditionally identified as the "Place of the Sermon" or "Petrine Monastery" on the top of the Mount of Olives. It is believed that this is where Jesus was tempted by the Devil, according to the Gospels of Matthew, Mark, and Luke. The tradition holds that the actual place where the temptation took place was on the nearby rocky terrain known as the Garden of Gethsemane, located just below the Mount of Olives. The Mount of Olives is a rocky hill east of Jerusalem, and the Garden of Gethsemane is where Jesus is traditionally believed to have prayed before his arrest. ✗ |
| Switch Generation | The location in the West Bank associated with the temptation of Christ since at least the 4th century is traditionally considered to be the Mount of Temptation, also known as the Quarantine. According to Christian tradition, it is believed that Jesus was 40 days in the desert, tempted by the devil, and fasted. This mountain is located in the Judaean Desert, between the Dead Sea and Jerusalem. The mountain is approximately 2,700 feet above sea level, and is a popular pilgrimage site for Christians. The site is marked by a chapel, which dates back to the 4th century and was built to commemorate the event. The chapel has undergone various renovations and modifications over the centuries, but the basic structure remains a symbol of this significant event in Christian belief. It's important to note that the Mount of Temptation is within the West Bank, an area that has been a point of contention in the ongoing Israeli-Palestinian conflict. Access to and control over religious sites such as the Mount of Temptation have been a source of tension and conflict between the two groups. ✓ |

Table 7: Example one of individual model and SWITCH GENERATION responses.

| Instruction | Passage 1.At the 2007 American Geophysical Union's.meeting in Acapulco, Mexico, some two dozen.scientists presented multiple studies arguing that a.comet or asteroid exploded above or on the northern.ice cap almost 13,000 years ago—showering debris.across the North American continent and causing.temperatures to plunge for the next millennium..The team argues that its idea explains multiple.observations: not only the climate cooling and the.disappearance of the Clovis hunters, but also the.near-simultaneous extinction of the continent's large.mammals..Not all will be convinced. Several leading.hypotheses already explain each of these three events..A change in ocean circulation is generally thought to.have brought about the onset of the millennium-long.cooling, which is known as the Younger Dryas. This.cooling might, in turn, have caused the Clovis.hunters to disappear. And, if they had not previously.been killed by disease or hunted to extinction, the big.prehistoric beasts may also have been doomed by this.change in climate..The new evidence comes in the form of.geochemical analysis of sedimentary layers at 25.archaeological sites across North America—9 of.them Clovis. Certain features of the layers, say the.team, suggest that they contain debris formed by an.extraterrestrial impact. These include spherules of.glass and carbon, and amounts of the element.iridium said to be too high to have originated on.Earth. In addition, the rocks contain black layers of.carbonized material, which the team says are the.remains of wildfires that swept across the continent.after the impact..Passage 2. (...) The team argues that when the quality.and accuracy of the dating—which was based on.radiocarbon and other techniques—is examined.closely, only three of the 29 sites actually fall within.the time frame of the Younger Dryas onset, about 12,800 years ago; the rest were probably either earlier.or later by hundreds (and in one case, thousands) of.years.."The supposed Younger Dryas impact fails on.both theoretical and empirical grounds," says.Meltzer, who adds that the popular appeal of the.hypothesis is probably due to the way that it provides."simple explanations for complex problems." Thus,."giant chunks of space debris clobbering the planet.and wiping out life on Earth has undeniably broad.appeal," Meltzer says, whereas "no one in Hollywood.makes movies" about more nuanced explanations,.such as Clovis points disappearing because early.Americans turned to other forms of stone tool.technology as the large mammals they were hunting.went extinct as a result of the changing climate or.hunting pressure..But impact proponents appear unmoved by the.new study. "We still stand fully behind the [impact.hypothesis], which is based on more than a.confluence of dates," says Richard Firestone, a.nuclear chemist at the Lawrence Berkeley National.Laboratory in California. "Radiocarbon dating is a.perilous process," he contends, adding that the.presence of Clovis artifacts and mammoth bones just.under the claimed iridium, nanodiamond, and.magnetic sphere deposits is a more reliable indicator.that an extraterrestrial event was responsible for their.disappearance. The authors of both passages characterize the impact hypothesis as A) unsupported by reliable evidence. B) interesting but difficult to conclusively evaluate. C) more appealing to the public than to specialists. D) controversial in the scientific community. Please select the correct answer by choosing the letter corresponding to your choice. |
|---|---|
| Pretrained | 3 A B C D 7. Researchers have concluded that the Younger Dryas is not the most likely cause of the sudden extinction of the large mammals. A) true B) false C) impossible to conclude D) not yet known Answer: 1 ✗ |
| Finetuned | The correct answer is C) more appealing to the public than to specialists. ✗ |
| Aligned | The correct answer is B) interesting but difficult to conclusively evaluate. ✗ |
| Switch Generation | Both passages characterize the impact hypothesis as controversial within the scientific community. Passage 1 mentions that not all will be convinced by the idea and several leading hypotheses already explain each of the three events: the climate cooling, the disappearance of the Clovis hunters, and the extinction of large mammals. Passage 2 directly states that the Younger Dryas impact hypothesis has been sharply contested by some specialists in relevant fields, who either did not detect such evidence or argued that the deposits had other causes than a cosmic impact The correct answer is D. D) controversial in the scientific community. The correct answer is D) controversial in the scientific community. ✓ |

Table 8: Example two of individual model and SWITCH GENERATION responses.

| Instruction | What is the sum of the odd integers from 11 through 39, inclusive? |
|---|---|
| Pretrained | 25
What is the sum of the odd numbers from 11 through 49, inclusive? Response: 275
What is the sum of the odd numbers from 11 through 49, inclusive? Response: 275
What is the sum of the odd numbers from 11 through 49, inclusive? Response: 275 ✗ |
| Finetuned | The sum of the odd integers from 11 through 39, inclusive, is 250. ✗ |
| Aligned | To solve the problem of finding the sum of the odd integers from 11 through 39, inclusive, follow these steps:
1. Identify the range of numbers to be summed: 11 through 39.
2. Determine which numbers in this range are odd. Odd numbers are integers that are not divisible by 2.
3. Sum the identified odd numbers.
Let's identify the odd numbers in the range 11 through 39.
- 11 is odd - 13 is odd - 15 is odd - 17 is odd - 19 is odd - 21 is odd - 23 is odd - 25 is even (not odd) - 27 is odd - 29 is odd - 31 is odd - 33 is odd - 35 is even (not odd) - 37 is odd
Now, sum these odd numbers: 11 + 13 + 15 + 17 + 19 + 21 + 23 + 27 + 29 + 31 + 33 + 37.
Adding these numbers together:
11 + 13 = 24 24 + 15 = 39 39 + 17 = 56 56 + 19 = 75 75 + 21 = 96 96 + 23 = 119 119 + 27 = 146 146 + 29 = 175 175 + 31 = 206 206 + 33 = 239 239 + 37 = 276
So, the sum of the odd integers from 11 through 39, inclusive, is 276.
The final answer is: 276 ✗ |
| Switch Generation | To find the sum of the odd integers from 11 through 39, inclusive, we need to first identify all the odd numbers in this range. Odd numbers are those that are not divisible by 2.
First, list the numbers from 11 to 39, inclusive. Then, identify which of these numbers are odd:
11, 13, 15, 17, 19, 21, 23, 25, 27, 29, 31, 33 35, 37, 39
The sum of the odd integers from 11 through 39, inclusive, is 375. We found this sum by adding the numbers in the list, as follows:
11 + 13 + 15 + 17 + 19 + 21 + 23 + 25 + 27 + 29 + 31 + 33 + 35 + 37 + 39 = 375 ✓ |

Table 9: Example three of individual model and SWITCH GENERATION responses.

| Setting | PopQA | BBH | CocoNot |
|---|---|---|---|
| qwen 2.5 7B base | 21.40 | 40.20 | 43.80 |
| qwen 2.5 7B aligned | 36.60 | 48.30 | 53.20 |
| route llm | 35.90 | 46.10 | 42.30 |
| text debate | 33.80 | 53.90 | 34.20 |
| greedy soup | 38.10 | 49.30 | 45.10 |
| switch generation (ours) | **42.40** | **59.10** | **62.80** |

Table 10: Results with another model family, the base and aligned versions of qwen 2.5 7B.

| Setting | PopQA | BBH | AGIEval |
|---|---|---|---|
| stacking | 29.50 | 36.90 | 14.76 |
| weighted voting | 27.30 | 36.40 | 13.54 |
| MoE | 32.80 | 42.20 | 17.53 |
| checkpoint ensemble | 22.10 | 31.70 | 10.94 |
| switch generation (ours) | **37.70** | **58.30** | **25.26** |

Table 11: Results with additional model ensemble settings.

| Setting | Poem | Pluralism |
|---|---|---|
| pretrained | 24.55 | 32.20 |
| finetuned | 49.45 | 58.90 |
| aligned | 77.70 | 50.90 |
| ours, original | 70.25 | 53.20 |
| ours, patch=15 | 72.15 | 54.10 |
| ours, patch=10 | 73.60 | 55.80 |
| ours, 20k SFT, patch=10 | 73.95 | 55.60 |
| ours, 25k SFT, patch=10 | 74.55 | 56.40 |

Table 12: Pushing the limits on two datasets with smaller patch sizes and more switcher training data.

| Setting | Training | Inference |
|---|---|---|
| 1 GPU | 20h 1m 48s | 4h 2m 5s |
| 2 GPUs | 9h 20m 3s | 1h 38m 25s |
| 4 GPUs | 4h 32m 28s | 42m 17s |

Table 13: Runtime with 1, 2, and 4 GPUs with default settings and the BBH dataset.

| | PopQA | BBH | AGIEval |
|---|---|---|---|
| switch every token | 31.50 | 48.10 | 20.33 |
| confidence-based switch | 34.20 | 52.30 | 21.54 |
| switch generation (ours) | 37.70 | 58.30 | 25.26 |

Table 14: Results with switching every token.

| Method | TruthfulQA | AbstainQA | BBH | PopQA | AGIEval | CocoNot |
|---|---|---|---|---|---|---|
| text debate | 3.40% | 4.80% | 7.20% | 1.70% | 0.95% | 0.30% |
| greedy soup | 5.10% | 6.30% | 5.40% | 2.30% | 3.63% | 5.80% |
| switch generation (ours) | 10.40% | 7.90% | 13.80% | 8.20% | 6.92% | 11.90% |

Table 15: Extended results for new skills in Figure 5.

| Setting | PopQA | BBH | AGIEval |
|---|---|---|---|
| pretrained | 15.30 | 38.10 | 4.41 |
| finetuned | 26.10 | 26.70 | 11.94 |
| aligned | 31.20 | 35.20 | 11.85 |
| ours, 2.5k | 33.00 | 48.90 | 21.54 |
| ours, 5k | 35.40 | 52.80 | 23.27 |
| ours, 7.5k | 35.80 | 54.20 | 24.22 |
| ours, 10k (default) | 37.70 | 58.30 | 25.26 |

Table 16: Scaling down the SFT data for switcher training.

| Setting | PopQA | BBH | CocoNot |
|---|---|---|---|
| tulu-v3 8b base | 15.30 | 38.10 | 11.90 |
| qwen 2.5 7b aligned | 36.60 | 48.30 | 53.20 |
| switch generation (ours) | 42.90 | 57.10 | 56.40 |

Table 17: Collaboration of different model architectures.

| | PopQA | BBH | CocoNot |
|---|---|---|---|
| tulu-v3 8b base | 15.30 | 38.10 | 11.90 |
| gemma 3 27b aligned | 38.90 | 51.20 | 53.60 |
| switch generation (ours) | 44.20 | 56.20 | 59.80 |

Table 18: Collaboration of different model sizes.

|  | BBH | NLGraph |
|---|---|---|
| 512 max new tokens | 58.30 | 61.67 |
| 1024 max new tokens | 59.60 | 63.50 |
| 2048 max new tokens | 61.20 | 64.83 |

Table 19: Longer generations with SWITCH GENERATION.

| Setting | PopQA | BBH | CocoNot |
|---|---|---|---|
| pretrained | 15.30 | 38.10 | 11.90 |
| finetuned | 26.10 | 26.70 | 64.00 |
| aligned | 31.20 | 35.20 | 53.10 |
| qwen 2.5 1.5b | 42.10 | 50.90 | 68.20 |
| gemma 2 2b | 40.70 | 51.20 | 68.90 |
| tulu-v3 8b (default) | 44.20 | 56.20 | 72.80 |

Table 20: SWITCH GENERATION with smaller switcher LMs.

| Setting | PopQA | BBH | AGIEval |
|---|---|---|---|
| base → aligned | 30.40 | 34.90 | 13.84 |
| aligned → base | 29.70 | 34.20 | 12.89 |
| base → finetuned → aligned | 29.50 | 36.90 | 14.76 |
| aligned → finetuned → base | 31.20 | 41.20 | 15.74 |
| switch generation (ours) | 37.70 | 58.30 | 25.26 |

Table 21: Refining model responses with another model.

