# OpenReview forum: "Don't Throw Away Your Pretrained Model"
_ICLR.cc/2026/Conference — ICLR 2026 Poster_

### Official Review · Reviewer_WS2y · 2025-10-28

**Soundness:** 2
**Presentation:** 3
**Contribution:** 3
**Rating:** 6
**Confidence:** 4

**Summary:**

The paper proposes to use a trained model selector to select from an aligned model and its pretrained version. The selection happens every few tokens, so the final response combines the capability of both models. The method improves over any single model, and is capable to answer some questions that both models cannot. Ablation study shows that the method can also be generalized to different models (instead of different versions of a single model).

**Strengths:**

1. The method is novel and provides valuable insights.
2. The paper provides thorough analysis of the results.

**Weaknesses:**

1. Some parts of the writing are unclear, e.g., in Table 3, we don't know the name of each model.
2. The size of the model selector seems to be the same as the pretrained and aligned models, which raises efficiency concerns. It would be nice to see either the result of a smaller selector model or an efficiency analysis.
3. Some experiment settings are unclear, and they might impact the fairness of the evaluation. See question 1.

**Questions:**

1. In the main experiment, has any of the baseline methods finetuned on these tasks (i.e., is the comparison with switch-g fair)? Has any of the baseline methods finetuned specifically for each of these tasks (i.e., is the comparison with switch-t fair)?
2. You mentioned that the switcher is a "small" model in section 2, but it's actually the same size of the pretrained and aligned models (8B), is there a reason for this?

---

> ### Author Response · Authors · 2025-11-17
> **Author Response**
>
> We would like to thank the reviewer for their thoughtful comments and feedback. Please find new experiments and proposed edits in the new version of the PDF, highlighted in blue.
>
> > Some parts of the writing are unclear, e.g., in Table 3, we don't know the name of each model.
>
> We extend the table caption to include more model descriptions:
>
> *“Performance when directly employing the trained switcher model for other model settings: PFA from another model family (qwen 2.5 7B), PA only (of Tulu-v3), PF and two versions of A (of Tulu-v3, one DPO and one RLVR), and three specialized LM experts (seperately fine-tuned versions of qwen 2.5 7b). The trained switcher consistently generalizes to these settings to varying extents.”*
>
> Now updated in the revised pdf.
>
> > The size of the model selector seems to be the same as the pretrained and aligned models, which raises efficiency concerns. It would be nice to see either the result of a smaller selector model or an efficiency analysis. You mentioned that the switcher is a "small" model in section 2, but it's actually the same size of the pretrained and aligned models (8B), is there a reason for this?
>
> In the model development lifecycle for any lab/company, the pretrained, finetuned, aligned LMs share the same size and architecture, because they are further-trained versions of each other. This is the setting we target in this work: model collaboration done in-house, with different model versions in the development lifecycle, without introducing external models. Thus it makes most sense to use one of the versions to initialize the switcher.
>
> However the methodology does not prevent using other LMs as the switcher. We try two extra:
>
> |                      | PopQA |  BBH  | CocoNot |
> |:--------------------:|:-----:|:-----:|:-------:|
> |      pretrained      | 15.30 | 38.10 |  11.90  |
> |       finetuned      | 26.10 | 26.70 |  64.00  |
> |        aligned       | 31.20 | 35.20 |  53.10  |
> |     qwen 2.5 1.5b    | 42.10 | 50.90 |  68.20  |
> |      gemma 2 2b      | 40.70 | 51.20 |  68.90  |
> | tulu-v3 8b (default) | 44.20 | 56.20 |  72.80  |
>
> Results show that 1) larger switcher LMs are better & 2) even for 1-2B switcher LMs, switch generation is still better than using any model individually.
>
> Now added as Table 20 in the updated pdf.
>
> > In the main experiment, has any of the baseline methods finetuned on these tasks (i.e., is the comparison with switch-g fair)? Has any of the baseline methods finetuned specifically for each of these tasks (i.e., is the comparison with switch-t fair)?
>
> Yes: the routellm, text collab, logit merge, proxy tuning, greedy soup, and dare ties baselines are trained/optimized/hyperparameter-tuned/… (because they need to) on each specific task, so it is fair comparison with switch-t (stronger than switch-g).

---

### Official Review · Reviewer_x8fX · 2025-10-31

**Soundness:** 3
**Presentation:** 3
**Contribution:** 2
**Rating:** 4
**Confidence:** 4

**Summary:**

This paper proposes SWITCH GENERATION, a collaborative inference framework that dynamically combines multiple model checkpoints (e.g., pretrained and aligned LMs) during response generation. Recognizing that alignment improves instruction following but harms creativity and calibration, the method uses a trained “switcher” LM to select, at each segment of the output, the model best suited for the current context. Evaluated on 18 tasks, it consistently outperforms individual models and existing collaboration baselines, demonstrating strong generalization, compositional reasoning, and efficient reuse of models typically discarded in training pipelines.

**Strengths:**

(1)Novel dynamic collaboration mechanism: Introduces fine-grained, trace-aware switching where models “take turns” generating response segments—more flexible than static ensembling or single-model inference.

(2)Effective and generalizable switcher design: The switcher LM is trained via supervised fine-tuning on rollout-based utility signals and generalizes well to unseen tasks and model combinations.

**Weaknesses:**

(1)The core idea of dynamically switching between models is not entirely new—it resembles prior work in other areas such as cloud-edge collaboration (e.g., ADASWITCH [1]) and LLM evaluation frameworks (e.g., Slide [2]). The paper does not sufficiently differentiate itself or provide deep mechanistic insights into why and when switching between base and aligned models works.

(2)The evaluation only uses same-sized (8B) checkpoints from the same model family (Tulu-v3). This fails to reflect practical scenarios where a large pretrained base model collaborates with a much smaller instruction-tuned model—a more common and resource-efficient deployment pattern.

(3)The paper does not compare against a simple but strong alternative: letting a capable base model refine or post-process the output of an aligned/SFT model. Such a baseline would better isolate the benefit of dynamic segment-level switching versus static post-hoc correction.

References:
[1]ADASWITCH: Adaptive Switching between Small and Large Agents for Effective Cloud-Local Collaborative Learning
[2]Slide: A framework integrating small and large language models for open-domain dialogues evaluation

**Questions:**

(1)Why not compare against a “refinement” baseline?
The paper should include a baseline where a strong pretrained (or larger) model processes or refines the output of an aligned/SFT model (e.g., by editing, re-ranking, or validating segments). This would help clarify whether the gains come from dynamic switching itself or simply from leveraging base-model capabilities in a post-hoc manner.


(2)Would the approach generalize to heterogeneous model scales or architectures?
The current evaluation assumes homogeneous model families and sizes. It remains unclear whether SWITCHGENERATION would work effectively when combining, say, a 70B base model with a 7B SFT model—or models from different families (e.g., Llama + Mistral). Have the authors tested or considered such settings?

---

> ### Author Response · Authors · 2025-11-17
> **Author Response**
>
> We would like to thank the reviewer for their thoughtful comments and feedback. Please find new experiments and proposed edits in the new version of the PDF, highlighted in blue.
>
> > The core idea of dynamically switching between models is not entirely new—it resembles prior work in other areas such as cloud-edge collaboration (e.g., ADASWITCH [1]) and LLM evaluation frameworks (e.g., Slide [2]).
>
> Thank you for the pointers! Our work, Switch Generation, is different from [1] and [2]:
>
> The “switching” in [1] is a multi-agent setup, where different models are responsible for generating different thoughts, actions, reflections. Every time a single model is selected to generate until EOS based on their role. In contrast, models in Switch Generation only generate part of the response in a sequence, not until EOS, for interleaved co-generation: this mid-generation model switching allows finer-grained control over where the skills of different models are most needed.
>
> [2] features an *evaluation* setting to select/combine the scores given by a large or small model, while Switch Generation targets a *generation* setting with multiple models.
>
> That said, [1-2] are very interesting ways of using multiple models in collaboration. **We now cite them in the related work section of the updated pdf.**
>
> > The paper does not sufficiently differentiate itself or provide deep mechanistic insights into why and when switching between base and aligned models works.
>
> On **differentiating our work**: we highlight how Switch Generation and the QTC problem is an extended and unique setting to the research question of routing in lines 118-122. To recap:
>
> “This differs from existing routing paradigms in that: 1) the trace $\mathbf{t} \neq \varnothing$; 2) each selected model $\mathbf{c}_i$ only generates text segments instead of the full response; 3) the switcher $f$ is applied multiple times instead of just once. This brings novel ways of combining model strengths, finer-grained control over model collaboration, and improved adaptability to diverse user requests.”
>
> On **why switching between base and aligned models works**: we highlight in lines 49-80 & Figure 1. To recap:
>
> “Since model responses are not monolithic and feature a wide variety of skills favoring different models (Figure 1), we propose Switch Generation, where different models in the training pipeline take turns to “speak” in a response sequence.”
>
> Expanding on this: instruction following might favor the finetuned model, recalling knowledge might favor the pretrained model, reasoning might favor the aligned model, so interleaved switch generation, where each model takes on different parts based on their strengths, is why this switching is necessary and works.
>
> > The evaluation only uses same-sized (8B) checkpoints from the same model family (Tulu-v3). This fails to reflect practical scenarios where a large pretrained base model collaborates with a much smaller instruction-tuned model—a more common and resource-efficient deployment pattern. Would the approach generalize to heterogeneous model scales or architectures? The current evaluation assumes homogeneous model families and sizes. It remains unclear whether SWITCHGENERATION would work effectively when combining, say, a 70B base model with a 7B SFT model—or models from different families (e.g., Llama + Mistral). Have the authors tested or considered such settings?
>
> In the model development lifecycle for any lab/company, the pretrained, finetuned, aligned LMs share the same size and architecture, because they are further-trained versions of each other. That is the setting we target in this work: model collaboration done in-house, without introducing external models.
>
> However the methodology of switch generation does not prevent models with different architectures/sizes and here are more experiments:
>
> Different model architectures:
>
> |                          |   PopQA   |    BBH    |  CocoNot  |
> |:------------------------:|:---------:|:---------:|:---------:|
> |      tulu-v3 8b base     |   15.30   |   38.10   |   11.90   |
> |    qwen 2.5 7b aligned   |   36.60   |   48.30   |   53.20   |
> | switch generation (ours) | **42.90** | **57.10** | **56.40** |
>
>
> Different model sizes:
>
> |                          |   PopQA   |    BBH    |  CocoNot  |
> |:------------------------:|:---------:|:---------:|:---------:|
> |      tulu-v3 8b base     |   15.30   |   38.10   |   11.90   |
> |    gemma 3 27b aligned   |   38.90   |   51.20   |   53.60   |
> | switch generation (ours) | **44.20** | **56.20** | **59.80** |
>
> Results show that Switch Generation is effective in both scenarios. Now added as Tables 17 and 18 in the updated pdf.

---

> > ### Author Response · Authors · 2025-11-17
> > **Author response, part 2**
> >
> > > The paper does not compare against a simple but strong alternative: letting a capable base model refine or post-process the output of an aligned/SFT model. Such a baseline would better isolate the benefit of dynamic segment-level switching versus static post-hoc correction.
> >
> > We experiment with the refine setting, where “a->b” means model b refine the output of model a, with the prompt: “You are given a question and a response from another language model. Please refine the response and generate a new one. Question: …\n Previous response: …\n New response:”
> >
> > |                              |   PopQA   |    BBH    |  AGIEval  |
> > |:----------------------------:|:---------:|:---------:|:---------:|
> > |        base -> aligned       |   30.40   |   34.90   |   13.84   |
> > |        aligned -> base       |   29.70   |   34.20   |   12.89   |
> > | base -> finetuned -> aligned |   29.50   |   36.90   |   14.76   |
> > | aligned -> finetuned -> base |   31.20   |   41.20   |   15.74   |
> > |   switch generation (ours)   | **37.70** | **58.30** | **25.26** |
> >
> > Results show that 1) refinement is helpful, 2) three-model refinement is better than two, 3) switch generation outperforms these settings.
> >
> > Please note that the refinement setting has higher inference costs than switch generation: k models need to each generate the (initial, refined) response of n tokens (assuming similar lengths of pre- and post-refined responses), so nk token budget in total; Switch Generation only generates n new tokens with each model contributing some.
> >
> > Now added as Table 21 in the updated pdf.
> >
> > All new experiments and proposed edits in our response are incorporated in the new version of the pdf.

---

### Official Review · Reviewer_ddHQ · 2025-11-01

**Soundness:** 3
**Presentation:** 3
**Contribution:** 3
**Rating:** 6
**Confidence:** 3

**Summary:**

This paper proposes the SWITCH GENERATION method to address the alignment tax problem in LLM alignment training. The core idea is that each model checkpoint has its advantages at different stages of the training process (pretrained/fine-tuned/aligned). Aligned models have strong inference capabilities, but pre-trained models are better at tasks such as creativity and calibration. The authors train a small switcher LM that dynamically selects which model to use to generate different text segments during generation (segment-level switching).

Experiments validated the method on 18 datasets, outperforming single models on 16/18 tasks, with an average improvement of 12.9% compared to 8 baselines. The method can generalize to unseen model combinations and tasks, and inference costs can be reduced through distillation (although only 57.5% of the gain is recovered).

Overall, this is an interesting task with a clear motivation, but the main problems are high computational costs and weak theoretical foundation.

**Strengths:**

1. **Important Problem, Strong Motivation:** Alignment tax is a real problem, supported by Table 1 and related literature. Pre-trained models do perform better on tasks such as WikiDYK and creativity. The idea of "not discarding pre-trained models" is quite meaningful given the high cost of training .

2. **Innovative Method Design:** Segment-level switching offers a new granularity option; using LM as a switcher leverages its language understanding capabilities to address the timing of switching, a reasonable design; model-labeled traces help the switcher learn the strengths of each model; and rollout evaluates the average performance of each choice.

3. **The experimental design is reasonable and comprehensive:** The 18 datasets are representative and cover different types (knowledge/reasoning/creativity/security); the 8 baselines cover different collaboration methods (API/text/logit/weight level); ablation experiments (patch size, training strategy) were conducted; the supplementary materials include statistical significance tests; and the generalization ability was also verified.

4. **Experimental results are good:** Significant improvements were observed on most tasks, outperforming single models on tasks 16 and 18, with an average improvement of 12.9% compared to the baseline.

5. **Strong generalization ability:** It can generalize to different model families (Qwen2.5), different numbers of models (2/4), specialized experts, and unseen tasks.

6. **Clear Writing:** Figure 1 effectively illustrates the motivation (the pretrained model does indeed perform better on some tasks), and Figure 2 fully demonstrates the methodology. Table 1 uses color to distinguish task types in the main results , facilitating understanding. The case studies (Tables 7-9) are compelling.

7. **High reproducibility:** We promise open-source code, data, and the switcher model. Appendix B contains detailed experimental information.

**Weaknesses:**

1. **High computational cost is the biggest problem.**
Inference requires n+1 models in memory (4 times the memory), and the switcher needs to be called every 50 tokens. Distillation reduces costs but only restores 57.5% of the gain. More importantly, the paper does not provide actual runtime and throughput comparisons, which are crucial for practical deployment. Hyperparameters are sensitive, and the patch size needs to be optimized for the specific task.

2. **Weak theoretical foundation**
Why is segment-level analysis superior to token-level or response-level analysis? There is a lack of theoretical explanation; the ablation in Table 2 is insufficient.
Why is tracing helpful? The R² in Figure 5 is 0.017, which shows almost no correlation.
- The definition of "new skills" (10.7%) in Table 5 is not rigorous enough and may be partly due to randomness.
How does the Switcher learn to recognize different skills? Figure 6 is only descriptive.
- The lack of in-depth analysis of these issues limits further improvement of the methodology.

3. **The difference between this method and routing methods is not significant enough.**
The essential difference lies mainly in the granularity and the use of traces; it is not a revolutionary breakthrough.

4. **Soundness-related issues**
Training the Switcher requires 10k samples × 32 rollouts (320k calls); sample efficiency was not discussed.
- Patch sizes vary greatly across different tasks (Table 2 shows 10-100), but there is no research on adaptive selection.
- The rule that the beginning and end must use aligned models is hard-coded and not elegant enough.

5. **Insufficient experimental analysis**
- Table 1 in the main text does not indicate statistical significance (although the supplementary materials do).
- Some datasets choose to be based on hypotheses rather than empirical evidence.
- Different evaluation indicators are not uniform
- On which tasks is switch generation inferior to single-model or baseline approaches? Why? These analyses are missing.

6. **Security Issues**
The Ethics Statement mentions that a misaligned base model may circumvent safety guardrails, but it lacks empirical assessment of the risks associated with harmful content generation. This is a serious concern.

7. **Generalization is still questionable.**
- Table 3 shows a significant decrease in generalization performance (setting 1: 5.8% → setting 4: 3.1%).
Can models with different architectures collaborate? For example, can GPT and LLaMA be combined ?
How did it perform with long text (>512 tokens)? Not evaluated.

8. **Minor Writing Issues**
The definitions of pretrained/finetuned/aligned in footnote 1 appeared relatively late. The R² of Figure 5 is 0.017, indicating extremely weak correlation; the statement of "positive correlation" in the text is overly optimistic.

**Questions:**

1. **What are the specific inference time, memory usage, and throughput?** This is crucial for evaluating usability. How does it perform with a single GPU/multiple GPUs?

2. Can the patch size be adjusted dynamically? Currently, we need to optimize for each task, which isn't very practical, is it?

3. Why use 8B as the switcher? Have you tried 1B? What was the bottleneck?

4. Can the training cost of 10k×32 rollouts be reduced? Have you tried importance sampling?

5. Where is the complete comparison of token-level switching? Table 2 only shows patch size, not token-level results.

6. Of the 10.7% "new skills" in Table 5, how much is a combination effect and how much is randomness? Can they be separated?

7. **How to ensure the security of the base model?** How to prevent harmful content? Can security constraints be added during switcher training?

8. Do long sequences (>512 tokens) still work?

9. Can models of different sizes (1B+7B+70B) collaborate? Wouldn't that be more practical?

10. In practical applications, how do you trade off performance and cost? Is a single model sufficient after distillation, or is multiple models necessary?

---

> ### Author Response · Authors · 2025-11-17
> **Author response**
>
> We would like to thank the reviewer for their thoughtful comments and feedback. Please find new experiments and proposed edits in the new version of the PDF, highlighted in blue.
>
> > Inference requires n+1 models in memory (4 times the memory), and the switcher needs to be called every 50 tokens. Distillation reduces costs but only restores 57.5% of the gain.
>
> Thank you for recognizing the distillation setting, where at inference time only one model is in memory, same as the cheap single-model baselines.
>
> We would like to highlight that “Restor(ing) 57.5% of the gain” is no small feat! This means an average of 13.26% relative improvement against the best single model averaged across Figure 3, just by having the models collaborate and distilling the collaboratively generated text back into a single model. The inference-time cost is the same while boosting utility by 13.26%: using “only” to describe this might be underselling what we achieved here.
>
> > More importantly, the paper does not provide actual runtime and throughput comparisons, which are crucial for practical deployment. How does it perform with a single GPU/multiple GPUs? In practical applications, how do you trade off performance and cost? Is a single model sufficient after distillation, or is multiple models necessary?
>
> Our implementation is by default using multiple GPUs for parallel model loading and text generation. It is also compatible with number of GPUs smaller than number of models: GPUs are then assigned to models in a round-robin way.
>
> We present the actual runtime under different switching granularity and GPU settings, with the BBH dataset and default hyperparameters:
>
> |        |  training  |  inference |
> |:------:|:----------:|:----------:|
> |  1 GPU | 20h 1m 48s |  4h 2m 5s  |
> | 2 GPUs |  9h 20m 3s | 1h 38m 25s |
> | 4 GPUs | 4h 32m 28s |   42m 17s  |
>
> Runtime mostly scales inverse linearly with the number of GPUs because of the round-robin assignment, with additional overhead due to loading in models to gpu vram.
>
> Now added as Table 13 in the updated pdf.
>
> It is always recommended to use multiple GPUs to run Switch Generation, for parallel generation. However, if only 1 GPU is available, the distillation setting then becomes super valuable as it only generates texts with a single model. Our work offers a spectrum of solutions for different levels of resourcefulness.
>
> > Why is segment-level analysis superior to token-level or response-level analysis? There is a lack of theoretical explanation; the ablation in Table 2 is insufficient. Where is the complete comparison of token-level switching? Table 2 only shows patch size, not token-level results.
>
> We compare with two switching-every-token settings: using the switcher LM for every single token or using token probability as a confidence measure and decide which tokens to switch (akin to [1-2]).
>
> |                          | PopQA |  BBH  | AGIEval |
> |:------------------------:|:-----:|:-----:|:-------:|
> |    switch every token    | 31.50 | 48.10 |  20.33  |
> |  confidence-based switch | 34.20 | 52.30 |  21.54  |
> | switch generation (ours) | 37.70 | 58.30 |  25.26  |
>
> Direct comparison shows that switching by patches is better than by token. Now added as Table 14 in the updated pdf.
>
> Switching and generating by patch resembles the setting in diffusion language models, and there lies a lot of theoretical and empirical work on its effectiveness [3-5].
>
> > Patch sizes vary greatly across different tasks (Table 2 shows 10-100), but there is no research on adaptive selection.
>
> A fixed patch size resembles the setting of diffusion language models (e.g. [4-5]), where fixed patch sizes never limited performance and utility. For now we treat patch size as a hyperparameter and make recommendations that it should be fine as long as it is not too large or small (Table 2: too large and the collaboration is sparse, too small and the switching is noisy), leaving adaptive selection as future work.
>
> > The R² of Figure 5 is 0.017, indicating extremely weak correlation; the statement of "positive correlation" in the text is overly optimistic.
>
> We remove the phrase “and there is a positive correlation between the two with R2 = 0.017.” in the updated pdf. We now only claim that “the most helpful sequences (with the highest treatment effect) are also among the most
> frequent”, which is true and more rigorous: the patterns with the highest frequency also have the highest treatment effect in Figure 5, across 2- and 3-length switching patterns.
>
> [1] Fei et al. "Nudging: Inference-time alignment of llms via guided decoding."
>
> [2] Shen et al. "Learning to decode collaboratively with multiple language models."
>
> [3] Karan and Du. "Reasoning with Sampling: Your Base Model is Smarter Than You Think."
>
> [4] Sahoo et al. "Simple and effective masked diffusion language models."
>
> [5] Arriola et al. "Block diffusion: Interpolating between autoregressive and diffusion language models."

---

> > ### Author Response · Authors · 2025-11-17
> > **Author response, part 2**
> >
> > > The definition of "new skills" (10.7%) in Table 5 is not rigorous enough and may be partly due to randomness. How does the Switcher learn to recognize different skills? Figure 6 is only descriptive.
> >
> > We reduce the randomness by running this analysis across more datasets and comparing against baselines: for what percentage of problems in the dataset, 1) none of the single models could answer correctly & 2) switch generation answers it correctly. We also compare this phenomenon with two strong baselines.
> >
> > |                          | TruthfulQA | AbstainQA |     BBH    |   PopQA   |  AGIEval  |   CocoNot  |
> > |:------------------------:|:----------:|:---------:|:----------:|:---------:|:---------:|:----------:|
> > |        text debate       |    3.40%   |   4.80%   |    7.20%   |   1.70%   |   0.95%   |    0.30%   |
> > |        greedy soup       |    5.10%   |   6.30%   |    5.40%   |   2.30%   |   3.63%   |    5.80%   |
> > | switch generation (ours) | **10.40%** | **7.90%** | **13.80%** | **8.20%** | **6.92%** | **11.90%** |
> >
> > Results show that 1) this is a consistent phenomenon, 2) switch generation is better at solving problems challenging for single models, with 9.85% $\pm$ 2.65% over the two baselines’ 3.06% $\pm$ 2.62% and 4.76% $\pm$ 1.50%.
> >
> > Now added as Table 15 in the updated pdf.
> >
> > > The difference between this method and routing methods is not significant enough. The essential difference lies mainly in the granularity and the use of traces; it is not a revolutionary breakthrough.
> >
> > We outline the three differences compared to traditional routing in lines 118-122: taking generation trace into consideration, the selected model only generates part of the response, the switching decision is made multiple times instead of once.
> >
> > Whether it is a “revolutionary breakthrough” is subjective, but switch generation and the QTC problem is at minimum a neat extension of routing that enables multiple models to collaborate in a more flexible and fine-grained manner.
> >
> > > Training the Switcher requires 10k samples × 32 rollouts (320k calls); sample efficiency was not discussed.
> >
> > If multiple GPUs are available, running the sampling/generation of different models in parallel is fast, see the previous results on runtime.
> >
> > We scale down switcher training by cutting the default 10k SFT examples to 7.5k, 5k, and 2.5k.
> >
> > |                     | PopQA |  BBH  | AGIEval |
> > |:-------------------:|:-----:|:-----:|:-------:|
> > |      Pretrained     | 15.30 | 38.10 |   4.41  |
> > |      Finetuned      | 26.10 | 26.70 |  11.94  |
> > |       Aligned       | 31.20 | 35.20 |  11.85  |
> > |      Ours, 2.5k     | 33.00 | 48.90 |  21.54  |
> > |       Ours, 5k      | 35.40 | 52.80 |  23.27  |
> > |      Ours, 7.5k     | 35.80 | 54.20 |  24.22  |
> > | Ours, 10k (default) | 37.70 | 58.30 |  25.26  |
> >
> > Results show that: better performance with more data & even the 2.5k switching setting is better than using any model individually.
> >
> > Now added as Table 16 in the updated pdf.
> >
> > > The rule that the beginning and end must use aligned models is hard-coded and not elegant enough.
> >
> > This is based on an empirical observation: in the beginning user instruction understanding is often needed & in the end summarizing the response is often needed. These requirements both favor the aligned model, thus our design choice.
> >
> > > The Ethics Statement mentions that a misaligned base model may circumvent safety guardrails, but it lacks empirical assessment of the risks associated with harmful content generation. How to prevent harmful content? Can security constraints be added during switcher training?
> >
> > We couldn’t show empirical evidence because what we described in the ethics statement is a “potential future risk” that is not happening currently: no pretrained base model is specifically trained to be “malicious”, “bad”, or “unsafe” by malicious actors, and frankly, we hope there will never be any.
> >
> > The idea of adding security constraints to switcher training is interesting though: a most straightforward approach could be at the start of generation, send the user instruction to some tool like the Google toxicity API first to get a safety score, then defer to the aligned model if there are risks.

---

> > > ### Author Response · Authors · 2025-11-17
> > > **Author response, part 3**
> > >
> > > > Table 3 shows a significant decrease in generalization performance (setting 1: 5.8% → setting 4: 3.1%). Can models with different architectures collaborate? For example, can GPT and LLaMA be combined ? Not evaluated. Can models of different sizes (1B+7B+70B) collaborate?
> > >
> > > In the model development lifecycle for any lab/company, the pretrained, finetuned, aligned LMs share the same size and architecture, because they are further-trained versions of each other. That is the setting we target in this work: model collaboration done in-house, without introducing external models.
> > >
> > > But the methodology of switch generation does not prevent models with different architectures/sizes.
> > >
> > > Different model architectures:
> > >
> > > |                          |   PopQA   |    BBH    |  CocoNot  |
> > > |:------------------------:|:---------:|:---------:|:---------:|
> > > |      tulu-v3 8b base     |   15.30   |   38.10   |   11.90   |
> > > |    qwen 2.5 7b aligned   |   36.60   |   48.30   |   53.20   |
> > > | switch generation (ours) | **42.90** | **57.10** | **56.40** |
> > >
> > >
> > > Different model sizes:
> > >
> > > |                          |   PopQA   |    BBH    |  CocoNot  |
> > > |:------------------------:|:---------:|:---------:|:---------:|
> > > |      tulu-v3 8b base     |   15.30   |   38.10   |   11.90   |
> > > |    gemma 3 27b aligned   |   38.90   |   51.20   |   53.60   |
> > > | switch generation (ours) | **44.20** | **56.20** | **59.80** |
> > >
> > > Results show that Switch Generation is effective in both scenarios. Now added as Tables 17 and 18 in the updated pdf.
> > >
> > > > How did it perform with long text (>512 tokens)?
> > >
> > > We increase the max_new_tokens for switch generation from 512 to 1024 and 2048 for two reasoning datasets, where this might be helpful.
> > >
> > > |                     |  BBH  | NLGraph |
> > > |:-------------------:|:-----:|:-------:|
> > > |  512 max new tokens | 58.30 |  61.67  |
> > > | 1024 max new tokens | 59.60 |  63.50  |
> > > | 2048 max new tokens | 61.20 |  64.83  |
> > >
> > > Results show that Switch Generation could effectively leverage the increased inference budget. Now added as Table 19 in the updated pdf.
> > >
> > > > The definitions of pretrained/finetuned/aligned in footnote 1 appeared relatively late.
> > >
> > > Moved it to page 1 in the updated pdf.
> > >
> > > > Why use 8B as the switcher? Have you tried 1B? What was the bottleneck?
> > >
> > > The setting we target in this work: model collaboration done in-house, with different model versions in the development lifecycle, without introducing external models. Thus it makes most sense to use one of the versions to initialize the switcher.
> > >
> > > However the methodology does not prevent using other LMs as the switcher. We try two extra:
> > >
> > > |                      | PopQA |  BBH  | CocoNot |
> > > |:--------------------:|:-----:|:-----:|:-------:|
> > > |      pretrained      | 15.30 | 38.10 |  11.90  |
> > > |       finetuned      | 26.10 | 26.70 |  64.00  |
> > > |        aligned       | 31.20 | 35.20 |  53.10  |
> > > |     qwen 2.5 1.5b    | 42.10 | 50.90 |  68.20  |
> > > |      gemma 2 2b      | 40.70 | 51.20 |  68.90  |
> > > | tulu-v3 8b (default) | 44.20 | 56.20 |  72.80  |
> > >
> > > Results show that 1) larger switcher LMs are better & 2) even for 1-2B switcher LMs, switch generation is still better than using any model individually.
> > >
> > > Now added as Table 20 in the updated pdf.

---

### Official Review · Reviewer_yuQM · 2025-11-02

**Soundness:** 3
**Presentation:** 4
**Contribution:** 3
**Rating:** 6
**Confidence:** 4

**Summary:**

This paper introduces Switch Generation, an inference-time model-collaboration method that trains a small “switcher” LM 𝑓 to decide which model checkpoint (pretrained, finetuned, aligned) should produce the next patch of tokens in a response. During switcher training the authors generate traces by randomly switching among candidate models, roll out 𝑘 continuations for each candidate, score those continuations with task-appropriate metrics, and turn the argmax choice into supervised labels for 𝑓. At inference time 𝑓 is queried every patch and selection is sampled with top-p. Experiments use three-stage Tulu-v3 checkpoints across 18 diverse datasets (QA, reasoning, creativity, safety, etc.), compare to 11 collaboration baselines, ablate patch size and switcher training, show generalization to unseen tasks and model pools, and demonstrate that distilling SWITCH GENERATION outputs back into a single aligned model recovers a substantial fraction of gains.

**Strengths:**

The paper proposes a clear, novel operationalization of the who speaks when problem that is conceptually simple and practically appealing: reuse existing checkpoints rather than discarding them. The method bridges routing and multi-agent text-level collaboration in a way that is easy to implement (small switcher LM + rollouts) and is evaluated comprehensively: many real tasks, multiple baseline families, ablations (patch size, untuned/random switcher), and analyses. The empirical gains are substantial and well aligned with the paper’s motivation. The distillation result is particularly useful for deployment trade-offs.

**Weaknesses:**

1. Experiments only use Tulu-v3. The authors can validate SWITCH GENERATION using model families beyond Tulu-v3 to show the approach is not specific to one model.
2. Because SWITCH GENERATION is conceptually similar to ensembles/routing, include direct comparisons to established ensemble methods (e.g. stacking, weighted voting, MOE and simple ensembling of checkpoints) and clarify where switch generation outperforms or trades off against those baselines.
3. For tasks where some metrics are worse than the original single-model baselines, please analyze why SWITCH GENERATION degrades performance on those metrics and whether tuning patch size, switcher regularization, or constrained candidate sets can recover or explain the losses.

**Questions:**

See  Weaknesses.

---

> ### Author Response · Authors · 2025-11-17
> **Author Response**
>
> We would like to thank the reviewer for their thoughtful comments and feedback. Please find new experiments and proposed edits in the new version of the PDF, highlighted in blue.
>
> > Experiments only use Tulu-v3. The authors can validate SWITCH GENERATION using model families beyond Tulu-v3 to show the approach is not specific to one model.
>
> We additionally employ the base and aligned versions of qwen 2.5 for collaboration.
>
> |                          |   PopQA   |    BBH    |  CocoNot  |
> |:------------------------:|:---------:|:---------:|:---------:|
> |     qwen 2.5 7b base     |   21.40   |   40.20   |   43.80   |
> |    qwen 2.5 7b aligned   |   36.60   |   48.30   |   53.20   |
> |         route llm        |   35.90   |   46.10   |   42.30   |
> |        text debate       |   33.80   |   53.90   |   34.20   |
> |        greedy soup       |   38.10   |   49.30   |   45.10   |
> | switch generation (ours) | **42.40** | **59.10** | **62.80** |
>
> Results show that Switch Generation continues to outperform both individual models and the three strong model collaboration baselines.
>
> Now added as Table 10 in the updated pdf.
>
> > Because SWITCH GENERATION is conceptually similar to ensembles/routing, include direct comparisons to established ensemble methods (e.g. stacking, weighted voting, MOE and simple ensembling of checkpoints) and clarify where switch generation outperforms or trades off against those baselines.
>
> We additionally compare with stacking (models sequentially refine the generation of the previous model), weighted voting (weighted by dev set performance), MoE (with top-1 expert activation), and simple ensemble of checkpoints (aka. Average soup [1]).
>
> |                          |   PopQA   |    BBH    |  AGIEval  |
> |:------------------------:|:---------:|:---------:|:---------:|
> |         stacking         |   29.50   |   36.90   |   14.76   |
> |      weighted voting     |   27.30   |   36.40   |   13.54   |
> |            MoE           |   32.80   |   42.20   |   17.53   |
> |    checkpoint ensemble   |   22.10   |   31.70   |   10.94   |
> | switch generation (ours) | **37.70** | **58.30** | **25.26** |
>
> Results show switch generation offers a better mode of collaboration, thanks to its flexibility in using models when their strengths are most needed.
>
> Now added as Table 11 in the updated pdf.
>
> > For tasks where some metrics are worse than the original single-model baselines, please analyze why SWITCH GENERATION degrades performance on those metrics and whether tuning patch size, switcher regularization, or constrained candidate sets can recover or explain the losses.
>
> We conduct additional experiments on Poem and Pluralism, two datasets where switch generation lagged behind single-model baselines. This is due to these tasks needing finer-grained control over the switching process, and we find that two factors would help on these two tasks: smaller patch sizes and more switcher training data. Compared to our default setting for these tasks (patch size=25, 10k switcher SFT data):
>
> |                         |  Poem | Pluralism |
> |:-----------------------:|:-----:|:---------:|
> |        pretrained       | 24.55 |   32.20   |
> |        finetuned        | 49.45 |   58.90   |
> |         aligned         | 77.70 |   50.90   |
> |      ours, original     | 70.25 |   53.20   |
> |      ours, patch=15     | 72.15 |   54.10   |
> |      ours, patch=10     | 73.60 |   55.80   |
> | ours, 20k SFT, patch=10 | 73.95 |   55.60   |
> | ours, 25k SFT, patch=10 | 74.55 |   56.40   |
>
> These changes narrow the performance gaps (7.45->3.15 and 5.70->2.50), thanks to scaling the granularity of switching decisions and enhancing switcher training. Though we are limited by time and resources in the rebuttal period, we believe this shows that Switch Generation can be scaled up for better performance and outperform these settings.
>
> Now added as Table 12 in the updated pdf.
>
> [1] Wortsman et al. "Model soups: averaging weights of multiple fine-tuned models improves accuracy without increasing inference time."

---

### Meta-Review · Area_Chair_cUWf · 2026-01-14

**Summary:**

The reviewers appreciated the strong motivation surrounding alignment and highlighted the segment-level switching mechanism as a novel approach to model collaboration. Empirical evaluations demonstrated clear performance gains over single models across several tasks. Upon examining the reviews and the rebuttal, we find that the primary concerns have been effectively addressed. Considering the value of combining pretrained and aligned models, we recommend acceptance.

**Reviewer Concerns:**

The main concerns are:
1) High computational and memory cost. The proposed method requires keeping several same size models in memory (or swapping them), which significantly increases inference resource requirements compared to standard deployment.
2) Evaluation and baselines.
3) The method is conceptually close to existing routing and ensemble paradigms.

Concern 1) was partially addressed in the rebuttal. While the authors proposed distillation as a solution, the runtime analysis provided in the rebuttal confirms that inference time scales inversely with the number of GPUs, which may still be costly in practice.

Concern 2) was addressed well by the new results provided in the rebuttal.

The authors also provided some explanations to address Concern 3). Although the proposed approach may be similar to existing paradigms, the AC and SAC believe that applying existing techniques to an ensemble of pretrained and aligned models is still an interesting study.

**Reviewer Scores:**

Reviewer x8fX (with rating 4) raised questions about baselines and evaluation settings. With the additional information and new results provided by the author, we think the reviewer may consider raising the score.
Other reviewers (with rating 6) will likely remain positive.

---

### Decision · Program_Chairs · 2026-01-26

Accept (Poster)